# Trends in tobacco, alcohol and branded fast-food imagery in Bollywood films, 1994-2013

**Ailsa J. McKay**[1], **Nalin Singh Negi**[2], **Nandita Murukutla**[2], **Anthony A. Laverty** [3]*,
**Pallavi Puri**[2], **Bella Vasant Uttekar**[4], **Sandra Mullin**[2], **Christopher Millett**[3,5]

**1** Department of Primary Care and Public Health, Imperial College London, London, United Kingdom, **2** Vital Strategies, New York, New York, United States of America, **3** Public Health Policy Evaluation Unit, School of Public Health, Imperial College London, London, United Kingdom, **4** Centre for Operations Research and Training, Vadodara, India, **5** Public Health Foundation of India, Gurugram, India

* a.laverty@imperial.ac.uk

## Abstract

### Background and aims

Exposure to tobacco, alcohol and fast-food use in films is associated with initiation of these behaviours. India is the world's largest film producer, but the extent of such imagery in Bollywood (Hindi cinema) films is unclear. We therefore aimed to describe the extent of and trends in tobacco, alcohol and fast-food imagery in Bollywood films, between 1994–2013.

### Methods

For the 15 top-grossing films each year between 1994–2013, the number of five-minute intervals containing product images were determined separately for tobacco, alcohol and fast-food. Both the proportion of films containing at least one image occurrence, and occurrences per film, were described overall and by year. Negative binomial regression described associations between film rating and occurrences/film, and estimated time-trends in occurrences/film, adjusted for rating.

### Results

We analysed 93 U-rated (unrestricted), 150 U/A-rated (parental guidance for children aged <12 years) and 55 A-rated (restricted to adult audience) films, containing 9,226 five-minute intervals (mean intervals/film 30.8, SD 4.0). 70% (n = 210), 93% (n = 278) and 21% (n = 62) of films contained at least one tobacco, alcohol and fast-food occurrence, respectively. Corresponding total mean occurrences/film were 4.0 (SD 4.9), 7.0 (4.7) and 0.4 (0.9). Tobacco occurrences were more common in U/A films (incidence rate ratio 1.49, 95% confidence interval 1.06–2.09) and A films (2.95; 1.95–4.48) than U-rated films. Alcohol occurrences were also more common in A-rated films than U-rated films (1.48; 1.15–1.85). Tobacco occurrences/film became less common over the observed period (adjusted trend -4% per annum; -2 to -7%; p <0.001), while alcohol (+2%; 0–3%; p = 0.02), and fast food (+8%; 2–14%; p = 0.01) occurrences/film became more common.

**Data Availability Statement:** All relevant data are within the paper and its Supporting Information files.

**Funding:** This work was supported by a Bloomberg Philanthropies grant to Vital Strategies (formerly the World Lung Foundation) and a Wellcome Trust Capacity Strengthening Strategic Award to the Public Health Foundation of India and a consortium of UK universities. CM and AL are funded by a NIHR Research Professorship awarded to CM. The Public Health Policy Evaluation Unit at Imperial College London is grateful for support from the NIHR School of Public Health Research. A portion of this work, including NSN, NM, PP and SM's time, were supported by a grant from Bloomberg Philanthropies to Vital Strategies (formerly the World Lung Foundation). The funders had no role in study design, data collection and analysis, decision to publish, or preparation of the manuscript.

**Competing interests:** The authors have declared that no competing interests exist.

## Conclusions

Although the extent of tobacco imagery in Bollywood films fell over 1994–2013, it is still frequently observed. Alcohol imagery is widespread, even in U-rated films, and trends in both alcohol and fast-food imagery are upwards.

## Introduction

Tobacco, alcohol and fast-food have been collectively described as health harmful commodities (HHCs) [1]. They are key contributors to many prevalent non-communicable diseases (NCDs) and important sources of morbidity and mortality worldwide [2]. Over recent years, associations between exposure to HHC imagery in films and HHC consumption have been observed across several geographical and cultural contexts [3–10]. In particular, key reports have highlighted the link between tobacco film imagery and youth tobacco use initiation [11, 12]. Whilst this association is relatively well-evidenced [3–6, 13] the high plausibility of a more generalisable principle and consistent evidence suggests such relationships hold for HHCs more generally [9, 10, 14, 15].

The intentional promotion of products via film placement is a form of indirect advertising. Indirect advertising aims to influence the feelings and attitudes of potential consumers towards products through methods other than the explicit recommendation applied in direct advertising. To-date, HHC advertising restrictions have again focused on tobacco more than other HHCs, and the relevance of indirect advertising has increased as constraints on direct advertising have tightened. Concern that direct advertising restrictions were stimulating indirect advertising led to a 2008 World Health Organization (WHO) clarification that product placement falls within the definition of advertising and promotion that the widely-adopted WHO Framework Convention on Tobacco Control (FCTC) recommends for constraint [16]. Further example of efforts applied to limit tobacco film placements include the 1998 Master Settlement Agreement, via which paid tobacco product placement became prohibited in the USA [17]. More recently, regulations introduced in 2012 in India require that strong justification be provided for any tobacco imagery in films. Moreover, if present, Government tobacco public service announcements and health disclaimers must be run when the film is shown, tobacco imagery must be blurred out, and health warnings shown whenever tobacco is displayed on screen [18].

With regard to alcohol and fast-food, recent examples of action taken to limit advertising include European Union directives aiming to coordinate alcohol advertising regulations across member states [19], and the focus on advertising and promotion in the UK Government's recent update of its childhood obesity strategy [20]. We are not aware that any interventions to limit alcohol or fast-food film imagery have been implemented to date, but recent global alcohol strategy recommendations acknowledge the relevance of both direct and indirect advertising, and the potential benefits of adopting marketing regulations based on the experience with tobacco [21–23].

As the various measures intended to limit direct and indirect advertising will interact with wider aspects of HHC control policies, public opinion and background trends in HHC use, it is difficult to quantify the impact of advertising regulations *per se*. Overall, recent evidence from top-grossing films in the USA suggests that while the proportion of films containing tobacco images decreased between 2010–2016, the number of tobacco images they collectively contained increased by 72% [24]. There is an important evidence gap relating to the extent of

HHC film imagery in the large, emerging markets, where the largest expansions in the HHC industries are anticipated [25], greater NCD burdens felt [26], and upward trends in NCD prevalence continue [27]. In general, tobacco use portrayal has been found to be more common in non-Hollywood versus Hollywood films [28]. and previous research suggests it is common in Bollywood (Hindi cinema) films: it was observed in approximately 50% of popular youth-rated Bollywood films in 2006–08 [29]. However, the extent of alcohol and fast-food portrayal–and trends in product appearances–are unclear. As India is the world's largest film producer [30], the world-leading film market (in terms of tickets sold) [31], and an increasingly important film exporter–as well as a strong proponent of restrictions on film-based tobacco imagery in particular–the Bollywood experience is of particular interest. As such we aimed here to describe:

1. The proportion of top-grossing Bollywood films that contain tobacco, alcohol and branded fast-food, and frequencies of product appearances per film

2. Trends in product appearances from 1994–2013

3. Associations between HHC product appearances per film and film rating, and interactions between film rating and trends in product appearances

## Methods

### Data collection

The 15 top-grossing Bollywood films were identified for each year between 1994 and 2013 via boxofficeindia.com (confirmed via Koimoi.com where information available). The top-grossing films were defined as those released during the relevant year with highest gross revenue from worldwide ticket sales at the date of data collection (2014). Where copies of these films were unavailable (one for each of 1995, 2002, 2003 and 2004; two in 1996), they were substituted with the next highest-grossing films (full list of films included available in S1 File). Central Board of Film Certification (CBFC) rating and film duration (i.e. run-time) information was obtained from the same sources. The CBFC ratings include 'unrestricted public exhibition' ('U'-rating), 'unrestricted public exhibition subject to parental guidance for children age <12 years' ('U/A'-rating) and 'restricted to adult [≥18 years] audiences' ('A'-rating). Whilst the Indian film certification process aims to ensure that no justification or glorification of either tobacco or alcohol use occurs in certified films, specific ratings are awarded on judgments made by the Board's view of the film in its entirety, and how this is typically influenced by any HHC portrayal is unclear.

The films were divided into 5-minute intervals for analysis. As per previous studies [32–35], we defined HHC image 'occurrences' as intervals containing at least one image of the product of interest. Thus multiple images within the same interval were coded as one occurrence, but a single image could be coded as two occurrences if it extended across the division between consecutive intervals. The main product types of interest were:

1. Tobacco: cigarette, bidi, pipe, cigar, chewing tobacco and/or tobacco for nasal inhalation

2. Alcohol: spirits, wine, beer and/or local alcohol products

3. Branded fast-food: any fast-food items in branded packaging (McDonalds, burger king, KFC, Pizza hut)

Occurrences/film were counted for each. Counts were also made separately for each of the different types of tobacco and alcohol product listed above (i.e. for occurrences of cigarette

images, bidi images, etc). Two coders each assessed an initial selection of 30 films, enabling assessment of inter-rater reliability. Cohen's kappa values were 0.90 (95% confidence interval (CI) 0.78–1.03; tobacco), 0.85 (0.71–0.98; alcohol) and 0.75 (0.59–0.91; fast-food). The remaining films were each reviewed by one of the coders who had assessed the initial film selection.

### Outcome variables

As per previous studies [32, 35], outcome variables described any (i.e. at least one) occurrence per film, and total number of occurrences per film, for each product type.

### Statistical analysis

We described film durations and rating distribution, by year. Univariable linear regression assessed association between film duration and year. The number of films exhibiting at least one product occurrence was described by year for each main product type, and mean occurrences per film per year described for both the main product types, and product subtypes. These outcomes were additionally described by rating category. As the counts of occurrences/film were over-dispersed, associations between rating and occurrences/film were investigated in negative binomial regression models, with and without adjustment for calendar year.

Negative binomial models, with and without adjustment for film rating, were used to investigate time-trends in occurrences/film using the 5 minute intervals as the unit of analysis, clustered at the movie level. In view of the information available, and limited relevance of more granular date information given the length of film production times, variability in release date by location, and expectation that film viewing would not be restricted to a particular time of year, 'time' was again inputted using calendar year alone. We assumed that this varied linearly with each of the outcomes as likelihood ratio tests did not provide evidence that the value of the likelihood function was improved by use of a categorical calendar year variable. We additionally considered interactions between calendar year and rating category. Estimated exposure effects are reported as incidence rate ratios (IRRs) denoting the proportional change in occurrences/film for a one unit increase in the relevant exposure. P-values were obtained from likelihood ratio tests.

## Results

### Film characteristics

Of the 300 films included, 93 (31.0%) were U-rated, 152 (50.7%) U/A-rated, and 55 (18.3%) A-rated. Mean (M) film duration was 154 minutes (standard deviation (SD) 20 minutes; range 100–240), and had reduced slightly over the observed period (regression coefficient = -1.2 minutes/year, 95% CI -0.78 to -1.5; p<0.0001; see S2 File). Total film duration equated to 9,226 intervals for analysis (M 30.8 per film, SD 4.0).

### Tobacco, alcohol and fast-food image occurrences

210 films (70%) contained at least one tobacco occurrence, 278 (93%) at least one alcohol occurrence, and 62 (21%) at least one branded fast-food occurrence. These outcomes are displayed by year in Fig 1, alongside mean occurrences per film per year (specific figures detailed in S3 File). Mean occurrences per film observed for the overall 1994–2013 period were 4.0 (SD 4.9) for tobacco, 7.0 (SD 4.7) for alcohol, and 0.4 (SD 0.9) for fast-food. Over the observed period, occurrences per film tended downwards for tobacco (across range 7.1 to 1.9 occurrences/film), and upwards for alcohol and fast-food (ranges 5.0–9.6 and 0–0.9, respectively). The majority of tobacco occurrences were of cigarettes, and spirits accounted for most alcohol

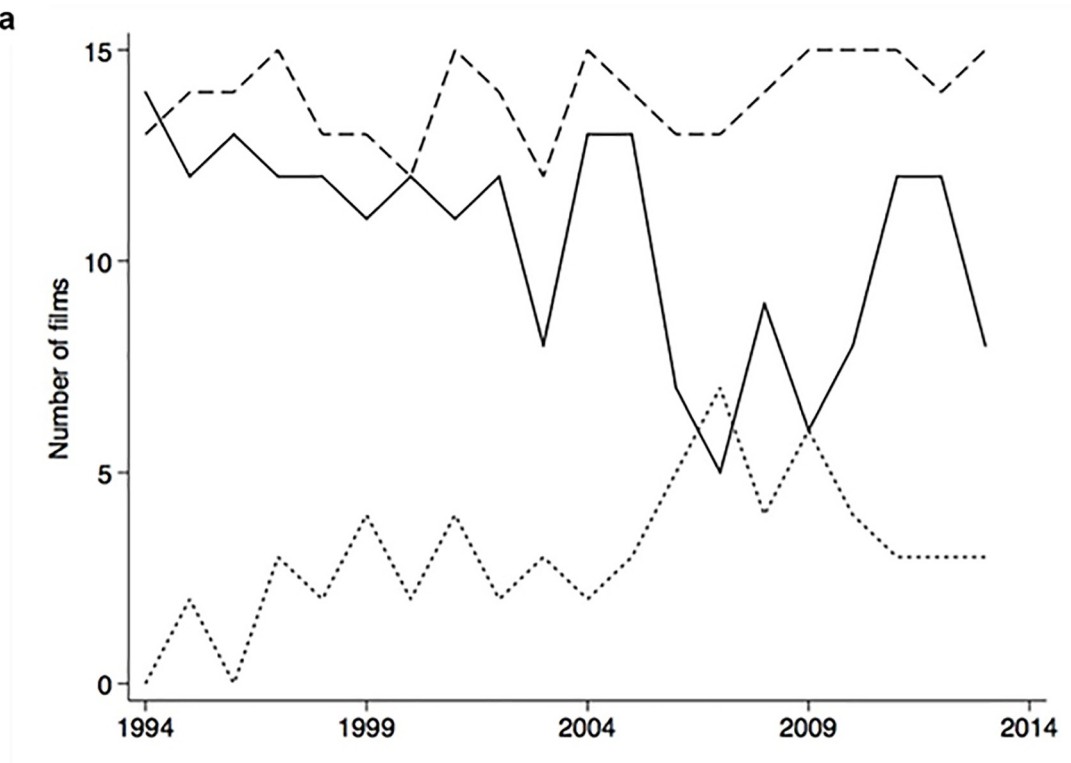

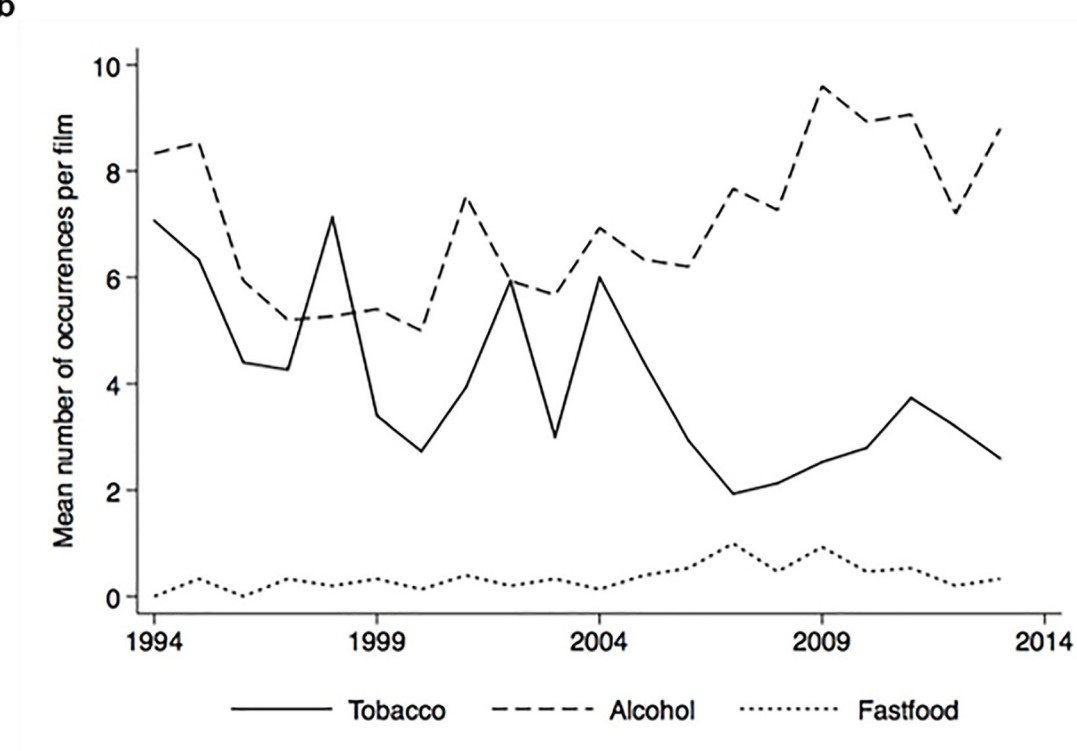

**Fig 1. Tobacco, alcohol and fast-food occurrences per film, by year. a**. Number of films containing any tobacco, alcohol or fast-food occurrence. **b**. Mean number of occurrences per film. NB. Specific figures for both outcomes detailed in S3 File.

**Table 1. Tobacco, alcohol and fast-food occurrences, by film rating category.** The number and proportion of films and 5-minute film intervals containing tobacco, alcohol and fast-food occurrences, are displayed, overall and by rating category.

| | | Films | | Intervals | |
|---|---|---|---|---|---|
| | | Total (n) | Containing ≥1 occurrence (n,%) | Total (n) | Containing ≥1 occurrence (n,%) |
| **Tobacco** | U | 93 | 56 (60.2) | 2,918 | 246 (8.4) |
| | U/A | 152 | 106 (69.7) | 4,694 | 540 (11.5) |
| | A | 55 | 48 (87.3) | 1,614 | 421 (26.1) |
| | Total | 300 | 210 (70.0) | 9,226 | 1,207 (13.1) |
| **Alcohol** | U | 93 | 83 (89.3) | 2,918 | 556 (19.1) |
| | U/A | 152 | 142 (93.4) | 4,694 | 1,079 (23.0) |
| | A | 55 | 53 (96.4) | 1,614 | 477 (29.6) |
| | Total | 300 | 278 (92.7) | 9,226 | 2,112 (22.9) |
| **Fast-food** | U | 93 | 22 (23.7) | 2,918 | 38 (1.3) |
| | U/A | 152 | 33 (21.7) | 4,694 | 477 (29.6) |
| | A | 55 | 7 (12.7) | 1,614 | 14 (0.9) |
| | Total | 300 | 62 (20.7) | 9,226 | 109 (3.5) |

U = unrestricted.

U/A = parental guidance for children aged <12 years).

A = restricted to adult audiences.

occurrences, although alcohol occurrences were of mixed compositions in later years (see S4 File).

The distribution of film ratings varied by year, with U/A more common than U- and A-rated films in recent years (see S5 File). Most films of all ratings contained at least one tobacco and at least one alcohol occurrence (see Table 1). We found strong evidence that both tobacco and alcohol occurrences were more common in films rated for older audiences (adjusted IRRs for U/A- and A- rated versus U- rated films 1.49 (95% CI 1.06–2.09) and 2.95 (1.95–4.48), respectively for tobacco, p <0.001; 1.13 (0.93–1.37) and 1.48 (1.15–1.85), respectively for alcohol, p = 0.01). Conversely, fast-food occurrences appeared relatively common in U- and U/A-rated films, but counts were low and we did not observe evidence of differences by rating on testing (adjusted IRRs for U/A- and A- rated versus U-rated films 0.81 (95% CI 0.43–1.52) and 0.61 (0.26–1.45), respectively, p = 0.53; see Table 2).

We found evidence for a downward trend in tobacco occurrences over the observed period (unadjusted and adjusted IRRs 0.96 (95% CI 0.93–0.98), p<0.001), and upward trends in alcohol (unadjusted and adjusted IRRs 1.02 (1.00–1.03), p = 0.02), and branded fast-food (unadjusted IRR 1.08 (1.02–1.13), p = 0.01; adjusted IRR 1.08 (1.02–1.14), p = 0.01) occurrences (see Table 3). We found no clear evidence of interaction between calendar year and film rating for any product type. Plots of occurrences/film per year are displayed by rating category in S6 File.

## Discussion

### Summary of findings

This study aimed to determine the extent of tobacco, alcohol and branded fast-food occurrences in the most widely-viewed Bollywood films of 1994–2013, and their trends over time. We found alcohol imagery to be highly prevalent (present in 93% of films, including 89% of U-rated films), with occurrences/film increasing at 2% per annum, to consistently >7 occurrences per film since 2007. Fast-food occurrences were the least frequently observed–in only seven films prior to 1999, although occurrences increased at 8% per year over the observed

**Table 2. Associations between health harmful commodity occurrences and film rating.** The incidence rate ratios describing associations between occurrences per film and film rating are displayed. n = 300 films in total (93 U-rated, 152 U/A-rated, 55 A-rated).

|  |  | Unadjusted IRR (95% CI) | p | Adjusted* IRR (95% CI) | p |
|---|---|---|---|---|---|
| **Tobacco** | **U** | ref | <0.001 | Ref | <0.001 |
|  | **U/A** | 1.34 (0.96–1.89) |  | 1.49 (1.06–2.09) |  |
|  | **A** | 2.89 (1.89–4.43) |  | 2.95 (1.95–4.48) |  |
| **Alcohol** | **U** | ref | 0.01 | Ref | 0.01 |
|  | **U/A** | 1.19 (0.98–1.44) |  | 1.13 (0.93–1.37) |  |
|  | **A** | 1.45 (1.14–1.84) |  | 1.48 (1.15–1.85) |  |
| **Fast-food** | **U** | ref | 0.56 | Ref | 0.53 |
|  | **U/A** | 0.92 (0.49–1.73) |  | 0.81 (0.43–1.52) |  |
|  | **A** | 0.62 (0.26–1.49) |  | 0.61 (0.26–1.45) |  |

IRR: incidence rate ratio; CI: confidence interval; *adjusted for year of film release.

U = unrestricted.

U/A = parental guidance for children aged <12 years).

A = restricted to adult audiences.

period. Tobacco occurrences were observed among 70% of films overall, with occurrences/film falling at 4% per year. Tobacco and alcohol occurrences were more frequently observed in films rated for older age-groups.

## Comparison with existing literature

A recent study of top-grossing Bollywood films from 2006–2008 reported at least one tobacco occurrence in 50% of U- and U/A- rated films, and 55% of U-, U/A- and A- rated films [29], and an older study found that 76% of a combination of Hindi, Tamil and Telugu language films released in 1992–2001 included tobacco imagery [36]. Despite some differences in the films analysed, these figures are broadly in keeping with our findings for these years (corresponding figures 42s.5% (U- and U/A-rated) and 47% (U-, U/A- and A-rated) for 2006–2008; 81% for 1994–2001). The 2006–2008 study used a definition of image occurrence ('the appearance of a tobacco product on screen') that differed from that we applied here (a five-minute film interval containing at least one tobacco product image), but the occurrences/film outcomes are similar, and followed the same patterning by film-rating. Our findings are also consistent with a more recent (2015–16) study that found that 51% of a combination of nationally- and locally- high-grossing films (mainly U- and U/A- rated) contained tobacco imagery [37]. The downward trend in tobacco occurrences we observed is in keeping with those in the US, Mexico and the UK in the 1990s-2000s [32, 38, 39].

The high prevalence of alcohol imagery in Bollywood films (even those of U-rating), is in keeping with findings from the US, Central and South America, and Europe [28, 38–40].

**Table 3. Time-trends in health harmful commodity occurrences.** The incidence rate ratios describing time-trends in tobacco, alcohol and fast-food occurrences, adjusted for film rating, are displayed. Unadjusted IRRs were as per the adjusted IRRs, except in the case of fast-food where the unadjusted IRR was 1.08 (95% CI 1.02–1.13). n = 300 films.

|  | Adjusted IRR (95% CI) | p |
|---|---|---|
| **Tobacco** | 0.96 (0.93–0.98) | <0.001 |
| **Alcohol** | 1.02 (1.00–1.03) | 0.02 |
| **Fast-food** | 1.08 (1.02–1.14) | 0.01 |

IRR: incidence rate ratio; CI: confidence interval.

Upward trends in alcohol imagery have not been observed elsewhere, but rates of imagery appearances have historically been relatively close to 100% as measured, limiting scope for upward movement. Fast-food film occurrences have not been extensively studied, although recent studies of top-grossing US films indicates that food retail establishment film appearances are usually fast-food establishments, and that food/drink brand placement is most common in films aimed at young people (PG13 rated) [41, 42].

## Strengths and limitations

This is the first study of tobacco, alcohol and fast-food imagery in Bollywood films [30, 31]. Strengths of this study include the use of standard methods to measure product occurrences, and data covering a 20-year period. Nonetheless, there are limitations that should be considered. In particular, our decision to record HHC occurrences within 5-minute intervals is consistent with several previous similar studies, but not all. Alternative outcome measures used by some include occurrences within scenes rather than intervals, and/or total duration of HHC on-screen visibility. We employed intervals rather than scenes due to the greater subjectivity and variability in scene definition. Measurement within intervals will also act as a proxy for total HHC on-screen durations to an extent. It is not known whether a particular outcome measure outperforms the others in terms of predicting initiation of HHC use, but our use of one particular outcome will somewhat limit comparability with some other studies. We were also unable to consider the *nature* of HHC images (e.g. which character used the product, and the character's response to product use), which has been demonstrated to affect impact [43]. And we were unable to estimate the number of impressions associated with each image (i.e. how often it was viewed) due to difficulty obtaining reliable average ticket price estimates for India over the relevant period. Whilst we considered all tobacco and alcohol use imagery, we included only branded fast-food imagery due to the greater difficulty and relatively high variability in definition of (unbranded) fast-food. Coding of alcohol occurrences where drinks were not branded relied on contextual factors, such as whether characters appeared under the influence of alcohol or whether this was mentioned by characters. Even with inclusion of only branded images, we found inter-rater reliability to be relatively low for fast-food, which may be due to the low levels of imagery identified. Our analyses could have benefitted from exploring this further, and from investigation of the specific tobacco and alcohol brands involved here, and further research could usefully examine which brands are responsible for the imagery in Bollywood films. Finally, we reviewed only a sample of 15 films released each year, based on their worldwide film grosses, as per data released by the relevant production companies. We do not know how the HHC content of these films compares with other high-grossing or lower-grossing Bollywood films, although recent research suggests tobacco imagery is more prevalent in local compared with Hindi or English language films [37]. Nonetheless, given our focus on the highest grossing films, many with large international audiences, our findings are still likely to have practical relevance.

## Implications for research and practice/policy

Over the time period covered by this study, several important measures relevant to HHC advertising were introduced in India. In 1995, a ban on cable television network tobacco and alcohol advertising was implemented [44]. India then anticipated many of the FCTC advertising restriction recommendations with implementation of the Cigarettes and Other Tobacco Products Act (COTPA) in 2004 [45]. Following an apparent subsequent increase in tobacco product and brand display in films [18], a 2005 COTPA amendment clarified that its definition of advertising and promotion included tobacco depictions in films [46]. Also regulations

requiring screening of anti-tobacco messages before and during on-screen tobacco displays, obscuring of product and brand imagery, and submission (to the CBFC) of editorial justification for any tobacco imagery, were introduced in 2012 [46].

Whilst we are unable to draw conclusions regarding the individual impacts of these specific measures from our study, comparisons between our observations relating to tobacco, and those for alcohol and fast-food, indicate that professional and public discussion of HHC advertising, its impact, and associated regulation may have had a role to play in reducing tobacco depictions in films. Such discussion and regulations have been much more pronounced for tobacco than for alcohol and fast-food, both in India and globally. More stringent laws from 2012 involved banning brand display and requiring lengthy anti-tobacco announcement. A recent study from Karnataka state indicated that compliance with these measures is low [37], and greater enforcement may therefore be of value. Nonetheless this measure has allowed the Government of India to have important screen time for these important public health messages, and further research on this intervention is needed [18].

Additional approaches found to have impact elsewhere–such as prohibition of funded product placement–could also help limit portrayal of all HHCs in Bollywood when combined with other measures [17]. Reinstating a requirement for all films with tobacco imagery to carry a particular (preferably 'A') rating, and instating the same for alcohol and fast-food, would likely have effect [18]. There is a varying picture of film subsidization across different Indian states and advancing recent recommendations to withhold state subsidies for films that portray HHC use may have additional impacts, although the extent of this is unclear [18]. Given the wide reach of Bollywood films these issues are important both in India and globally and coordinated policies are urgently needed to stem the rise in on-screen HHC imagery. Finally, previous research in India as elsewhere also highlights that when advertising is restricted in one medium, it is likely to move to another, and so governments should consider all advertising, promotion and sponsorship and ensure comprehensive implementation of FCTC article 13 [18].

## Conclusions

Alcohol and tobacco imagery is widespread in Bollywood films, and branded fast-food imagery increasingly so. Alcohol and fast-food film content may benefit from regulation as for tobacco. Additional measures such as prohibition of funded product placement, and of HHC portrayal in youth-rated films, could help further limit exposure of audiences to these images and their negative health impacts.

## Supporting information

**S1 File. Films included in analysis.**
(DOCX)

**S2 File. Mean film duration by year.**
(DOCX)

**S3 File. Image occurrences by year.** Proportion of films containing any tobacco, alcohol or fast-food image occurrence, and mean number of occurrences per film, by year. n = 15 films per year, 300 in total.
(DOCX)

**S4 File. Occurrences of tobacco and alcohol subtypes per film, by year.** a. Mean number of occurrences per film for tobacco sub-types. b. Mean number of occurrences per film for

alcohol sub-types.
(DOCX)

**S5 File. Distribution of film ratings, by year.**
(DOCX)

**S6 File. Tobacco, alcohol and fast-food occurrences/film per year, stratified by film rating.**
Mean number of tobacco (a), alcohol (b), and fast-food (c) occurrences/film per year, by rating
category. Numbers of films per category per year (d).
(DOCX)

**S1 Dataset.**
(RAR)

## Author Contributions

**Conceptualization:** Ailsa J. McKay, Nandita Murukutla, Anthony A. Laverty, Christopher
Millett.

**Data curation:** Ailsa J. McKay.

**Funding acquisition:** Nandita Murukutla, Christopher Millett.

**Investigation:** Anthony A. Laverty, Christopher Millett.

**Methodology:** Christopher Millett.

**Supervision:** Anthony A. Laverty.

**Writing – original draft:** Ailsa J. McKay.

**Writing – review & editing:** Nalin Singh Negi, Nandita Murukutla, Anthony A. Laverty, Pal-
lavi Puri, Bella Vasant Uttekar, Sandra Mullin, Christopher Millett.

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
