## [Decision Letter · Decision Letter 0]

5 Sep 2019

PONE-D-19-20655

Trends in tobacco, alcohol and branded fast-food imagery in Bollywood films, 1994-2013

PLOS ONE

Dear Dr Laverty,

Thank you for submitting your manuscript to PLOS ONE. After careful consideration, we feel that it has merit but does not fully meet PLOS ONE’s publication criteria as it currently stands. Therefore, we invite you to submit a revised version of the manuscript that addresses the points raised during the review process.

We would appreciate receiving your revised manuscript by Oct 20 2019 11:59PM. To enhance the reproducibility of your results, we recommend that if applicable you deposit your laboratory protocols in protocols.io, where a protocol can be assigned its own identifier (DOI) such that it can be cited independently in the future. For instructions see: http://journals.plos.org/plosone/s/submission-guidelines#loc-laboratory-protocols

We look forward to receiving your revised manuscript.

Kind regards,

Stanton A. Glantz

Academic Editor

PLOS ONE

Additional Editor Comments:

In addition to responding to the reviewers, the authors need to include an assessment of the effects of India's strengthened rules limiting smoking in movies, requiring warnings before and during films, and other policies to complete the analysis in this paper.

"CM and AL are funded by a NIHR Research Professorship awarded to CM. The Public Health Policy Evaluation Unit at Imperial College London is grateful for support from the NIHR School of Public Health Research. A portion of this work, including NSN, NM, PP and SM’s time, were supported by a grant from Bloomberg Philanthropies to Vital Strategies (formerly the World Lung Foundation)."

"This work was supported by a Bloomberg Philanthropies grant to Vital Strategies (formerly the World Lung Foundation) and a Wellcome Trust Capacity Strengthening Strategic Award to the Public Health Foundation of India and a consortium of UK universities.  The funders played no role in the design, conduct or reporting of the study."

Please provide an amended Funding Statement that declares *all* the funding or sources of support received during this specific study (whether external or internal to your organization) as detailed online in our guide for authors at http://journals.plos.org/plosone/s/submit-now.  

Please state what role the funders took in the study.  If any authors received a salary from any of your funders, please state which authors and which funder. If the funders had no role, please state: "The funders had no role in study design, data collection and analysis, decision to publish, or preparation of the manuscript."

Reviewers' comments:

Reviewer's Responses to Questions

**Comments to the Author**

1. Is the manuscript technically sound, and do the data support the conclusions?

Reviewer #1: Yes

Reviewer #2: Yes

2. Has the statistical analysis been performed appropriately and rigorously? 

Reviewer #1: Yes

Reviewer #2: Yes

3. Have the authors made all data underlying the findings in their manuscript fully available?

Reviewer #1: Yes

Reviewer #2: No

4. Is the manuscript presented in an intelligible fashion and written in standard English?

Reviewer #1: Yes

Reviewer #2: Yes

5. Review Comments to the Author

Reviewer #1: The aim of this study is to analyze the proportion of top-grossing Bollywood films that contain tobacco, alcohol and branded fast-food, and to describe trends in product appearance over time. Furthermore, associations between the occurrence of tobacco, alcohol and branded fast-food , and to film ratings were analyzed.

Previous research has shown, that smoking in movies is strongly correlated with smoking initiation in young people. Given the fact, that India is the world’s largest film producer, and the world-leading film market the research questions of this manuscript are important for public health in India and beyond.

Overall, this manuscript is well written, and worth to be published in a major journal.

Strengths include:

1. A large sample size of films (N=300)

2. A long observational period (2 decades)

3. A standard method, which has been used for the content coding.

Limitations of the research are fairly discussed.

I have the following minor suggestions:

1. WHO published several recommendations to tackle the issue of smoking in films [World Health Organisation. Smoke-free movies: from evidence to action. Third edition. WHO, 2015.], which could be discussed in detail in the policy section of the manuscript.

2. Being not familiar with Bollywood films, how are these films financed? Is there any tax payer money in it?

3. Tables: the abbreviations “U”, “U/A”, and “A” should be explained.

Reviewer #2: COMMENTS TO THE AUTHOR

SUMMARY

This manuscript reports the results from a content analysis assessing tobacco, alcohol, and branded fast food in popular Bollywood movies. The sample frame includes the top 15 box office hits for a 20 year period ending in 2013. Films were assessed by determining if tobacco, alcohol or branded fast food appeared in 5 minute intervals, with unbranded tobacco and alcohol appearances counting. The analysis examined tobacco, alcohol, and fast food appearances by movie rating and year of release. The results found little fast food promotion in Bollywood movies, universal alcohol appearance and frequent tobacco. More tobacco was found in more adult-rated movies. Over time there was a decline in tobacco, a slight upward trend in alcohol and an upward trend for fast food.

GENERAL COMMENTS

This is an excellent study that can be useful from a policy perspective for regulators in India. I have only a few comments.

MAJOR COMMENTS

Content coding: It would be good for you to more clearly define what you mean by branded fast food. Would these be quick service restaurants or would this include convenience foods like breakfast cereals and sodas? Would this be international brands only—like KFC—or would this include (if there are any) regional and national Indian brands? It would be helpful to have a table that lists brands responsible for the placements.

Interrater reliability of fast food was probably low, in part, because of the low prevalence of the practice.

With respect to alcohol, how did you ascertain if an unbranded drink was alcohol?

Analysis: It isn’t clear to me from my reading of the methods if the trend analysis was conducted at the level of the 5 minute interval or the movie. Please make that clear. If any analysis occurs at the 5 minute interval level, it needs to account for clustering at the movie level.

Results:

Excluding the first year, the frequency of alcohol occurrences almost doubled over the period. It would be helpful to understand what brands were displayed in order to ascertain whether this was due to national brands or international brands. Alcohol companies make no secret about placement deals with the movie industry, and it’s important to establish what company is responsible for the placements. For example, in the US, Budweiser is responsible for about one-fifth of placements. I hope the content analysis captured these data.

MINOR COMMENTS

Page Lines Comment

7 Para 2 I don’t think rating is numeric so you can’t say “as rating increased”

9 Para 1 “Out analyses” should be “Our analyses”

6. PLOS authors have the option to publish the peer review history of their article (what does this mean?). If published, this will include your full peer review and any attached files.

Reviewer #1: No

Reviewer #2: Yes: James D Sargent

---

## [Author Response · Author response to Decision Letter 0]

19 Feb 2020

We thank the editor and reviewers for their consideration of this paper and their detailed comments. We feel that the paper is now much improved and provide specific responses to points below

Editor Comments

In addition to responding to the reviewers, the authors need to include an assessment of the effects of India's strengthened rules limiting smoking in movies, requiring warnings before and during films, and other policies to complete the analysis in this paper.

We thank the editor for this comment and we have now expanded our discussion of the 2012 measures at the end of the Discussion section, drawing in part on the WHO report suggested by reviewer 2. We point out that this measure has allowed valuable screen time to be used for public health messaging. Nonetheless existing evidence does suggest compliance may be low and further examination is required. However, as our available data only includes one data point after the implementation of these measures, we are unable to provide statistical analysis of this policy.

We are happy to share this data on request 

"CM and AL are funded by a NIHR Research Professorship awarded to CM. The Public Health Policy Evaluation Unit at Imperial College London is grateful for support from the NIHR School of Public Health Research. A portion of this work, including NSN, NM, PP and SM’s time, were supported by a grant from Bloomberg Philanthropies to Vital Strategies (formerly the World Lung Foundation)."

"This work was supported by a Bloomberg Philanthropies grant to Vital Strategies (formerly the World Lung Foundation) and a Wellcome Trust Capacity Strengthening Strategic Award to the Public Health Foundation of India and a consortium of UK universities. The funders played no role in the design, conduct or reporting of the study."

a. Please provide an amended Funding Statement that declares *all* the funding or sources of support received during this specific study (whether external or internal to your organization) as detailed online in our guide for authors at http://journals.plos.org/plosone/s/submit-now. 

We apologise for this initial inconsistency. We have now amended the text and moved all funding information from the Acknowledgements to the funding section. The whole funding statement now reads:

This work was supported by a Bloomberg Philanthropies grant to Vital Strategies (formerly the World Lung Foundation) and a Wellcome Trust Capacity Strengthening Strategic Award to the Public Health Foundation of India and a consortium of UK universities. CM and AL are funded by a NIHR Research Professorship awarded to CM. The Public Health Policy Evaluation Unit at Imperial College London is grateful for support from the NIHR School of Public Health Research. A portion of this work, including NSN, NM, PP and SM’s time, were supported by a grant from Bloomberg Philanthropies to Vital Strategies (formerly the World Lung Foundation). The funders had no role in study design, data collection and analysis, decision to publish, or preparation of the manuscript. 

b. Please state what role the funders took in the study. If any authors received a salary from any of your funders, please state which authors and which funder. If the funders had no role, please state: "The funders had no role in study design, data collection and analysis, decision to publish, or preparation of the manuscript."

We have now amended the funding statement to read "The funders had no role in study design, data collection and analysis, decision to publish, or preparation of the manuscript” as requested 

Thank you. We have now done this

We have now added this to the end of the manuscript as requested

Reviewer 1 comments

The aim of this study is to analyze the proportion of top-grossing Bollywood films that contain tobacco, alcohol and branded fast-food, and to describe trends in product appearance over time. Furthermore, associations between the occurrence of tobacco, alcohol and branded fast-food , and to film ratings were analyzed. Previous research has shown, that smoking in movies is strongly correlated with smoking initiation in young people. Given the fact, that India is the world’s largest film producer, and the world-leading film market the research questions of this manuscript are important for public health in India and beyond.

Overall, this manuscript is well written, and worth to be published in a major journal. 

Strengths include:

1. A large sample size of films (N=300)

2. A long observational period (2 decades)

3. A standard method, which has been used for the content coding.

Limitations of the research are fairly discussed.

We thank the reviewer for their detailed reading and positive assessment of this manuscript.

I have the following minor suggestions:

1. WHO published several recommendations to tackle the issue of smoking in films [World Health Organisation. Smoke-free movies: from evidence to action. Third edition. WHO, 2015.], which could be discussed in detail in the policy section of the manuscript.

We thank the reviewer for this point and we now mention in the implications for policy section that governments need to move towards full implementation of FCTC article 13, as recommended in the WHO report. 

2. Being not familiar with Bollywood films, how are these films financed? Is there any tax payer money in it?

The reviewer makes an interesting point. Financing of films is complex, being funded by private production houses, although different Indian states do give incentives which vary across states. We now mention this in the Discussion, specifically that the use of such incentives could be amended for films which contain imagery analysed here. 

3. Tables: the abbreviations “U”, “U/A”, and “A” should be explained.

We have now added explanations to these abbreviations 

Reviewer 2 comments

SUMMARY

This manuscript reports the results from a content analysis assessing tobacco, alcohol, and branded fast food in popular Bollywood movies. The sample frame includes the top 15 box office hits for a 20 year period ending in 2013. Films were assessed by determining if tobacco, alcohol or branded fast food appeared in 5 minute intervals, with unbranded tobacco and alcohol appearances counting. The analysis examined tobacco, alcohol, and fast food appearances by movie rating and year of release. The results found little fast food promotion in Bollywood movies, universal alcohol appearance and frequent tobacco. More tobacco was found in more adult-rated movies. Over time there was a decline in tobacco, a slight upward trend in alcohol and an upward trend for fast food.

We thank the reviewer for their detailed reading and consideration of this paper.

GENERAL COMMENTS

This is an excellent study that can be useful from a policy perspective for regulators in India. I have only a few comments.

Thank you

MAJOR COMMENTS

Content coding: It would be good for you to more clearly define what you mean by branded fast food. Would these be quick service restaurants or would this include convenience foods like breakfast cereals and sodas? Would this be international brands only—like KFC—or would this include (if there are any) regional and national Indian brands? It would be helpful to have a table that lists brands responsible for the placements.

We classified branded fast foods as those from large international chains such as McDonalds, burger king, KFC, Pizza hut, and we now clarify this in the manuscript.

Interrater reliability of fast food was probably low, in part, because of the low prevalence of the practice.

The reviewer makes a very insightful point here and we now mention that this low Interrater reliability may be due to this reason in the strengths and limitations section. 

With respect to alcohol, how did you ascertain if an unbranded drink was alcohol?

We used a coding sheet which noted whether there were occurrences of drinking Whisky; Vodka, Rum, Wine, Beer, Champagne, Desi or unbranded alcohol. Classification of unbranded alcohol depended on the context of the scene and the dialogue at that time. For example, if a person is visibly under the influence of alcohol or this is being discussed, and they are drinking, it is assumed that this is unbranded alcohol. We now clarify this in the manuscript

Analysis: It isn’t clear to me from my reading of the methods if the trend analysis was conducted at the level of the 5 minute interval or the movie. Please make that clear. If any analysis occurs at the 5 minute interval level, it needs to account for clustering at the movie level.

We thank the reviewer for pointing this out and now clarify this in the methods. We present descriptive statistics at both the 5 minute level and the movie level, with analyses using the 5 minute interval data. These analyses clustered at the movie level.

Results:

Excluding the first year, the frequency of alcohol occurrences almost doubled over the period. It would be helpful to understand what brands were displayed in order to ascertain whether this was due to national brands or international brands. Alcohol companies make no secret about placement deals with the movie industry, and it’s important to establish what company is responsible for the placements. For example, in the US, Budweiser is responsible for about one-fifth of placements. I hope the content analysis captured these data.

The reviewer makes a very interesting point here and we agree that it would be very informative to know which brands specifically were involved here. Unfortunately, due to time and resource constraints, our data does not contain this information. However, as we agree with the reviewer that this is an important issue we have now added to the strengths and limitations section that this information would be a useful addition to future research. 

MINOR COMMENTS

7 Para 2 I don’t think rating is numeric so you can’t say “as rating increased”

Thank you, we have now changed this to “in films rated for older audiences”

9 Para 1 “Out analyses” should be “Our analyses”

 Thank you – we have now corrected this

---

## [Editor Report · Decision Letter 1]

21 Feb 2020

Trends in tobacco, alcohol and branded fast-food imagery in Bollywood films, 1994-2013

PONE-D-19-20655R1

Dear Dr. Laverty,

We are pleased to inform you that your manuscript has been judged scientifically suitable for publication and will be formally accepted for publication once it complies with all outstanding technical requirements.

With kind regards,

Stanton A. Glantz

Academic Editor

PLOS ONE
---

## [Editor Report · Acceptance letter]

15 May 2020

PONE-D-19-20655R1 

Trends in tobacco, alcohol and branded fast-food imagery in Bollywood films, 1994-2013 

Dear Dr. Laverty:

I am pleased to inform you that your manuscript has been deemed suitable for publication in PLOS ONE. Congratulations! Your manuscript is now with our production department. 

With kind regards,

on behalf of

Professor Stanton A. Glantz 

Academic Editor

PLOS ONE